# Utilisation Potential of Mechanical Material Loads during Grinding by Means of a Novel Tool Concept

**Marco Eich** [1,2,*], **Daniel Meyer** [1,2] and **Carsten Heinzel** [1,2,*]

1   Leibniz Institute for Materials Engineering IWT, Badgasteiner Straße 3, 28359 Bremen, Germany; dmeyer@iwt.uni-bremen.de
2   MAPEX Center for Materials and Processes, FB4 University of Bremen, Bibliothekstr. 1, 28359 Bremen, Germany
*   Correspondence: eich@iwt.uni-bremen.de (M.E.); heinzel@iwt.uni-bremen.de (C.H.); Tel.: +49-421-218-51182 (M.E.)

**Abstract:** The objective of this work is to improve the surface and subsurface properties of steel parts by means of a new grinding tool concept featuring nearly spherical grains in an elastic bonding system and to uncover the underlying mechanisms leading to the intended improvement of surface integrity. The resulting workpiece topography and subsurface properties, such as residual stresses, are evaluated to characterise and assess the potential of this novel tool concept. Micrographs and EBSD images are also analysed. The results show increased mechanical process loads and resulting favourable subsurface properties in terms of mechanically induced plastic deformation and compressive residual stresses, revealing the high potential of spherical grains in an elastic bonding system.

**Keywords:** grinding; surface integrity; tool concept

## 1. Introduction

The ever-increasing demands of industry considering the functional properties of manufactured components, and therefore the surface and subsurface properties, require a rethinking of traditional finishing processes. Grinding is typically used as a finishing process and is characterised by a thermomechanical load on the workpiece surface [1–3]. In view of the potentially negative impact of the thermal effects on subsequent functional properties, particular attention is generally paid to reducing the thermal load.

### 1.1. Size Effect

In addition, various process strategies can be used to favour the proportion of mechanical loading in the machining process and thus have a beneficial effect on the resulting surface integrity. According to Heinzel et al., there is a correlation between the specific related grinding power $P_{c''}$ at small chip thicknesses $h_{cu}$ and the mechanical load in the grinding process [4]. As the chip thickness is reduced, there is a size effect in the specific grinding energy which increases exponentially. In this case, the removal mechanism shifts towards micro-ploughing. By taking this mechanism into account, it is possible to select suitable process parameters and exploit this effect [4,5]. In their investigations, Borchers et al. [6] used the size effect in external cylindrical plunge grinding of steel with low cutting speeds in the range of $v_c = 1$ m/s. The result was an increase in compressive residual stresses with a depth effect of approximately 50 µm due to higher mechanical process loads on the material. Grain refinement and dislocations leading to compressive residual stresses were identified as the effective mechanism by EBSD measurements [5,6].

### 1.2. Elastic Bonding System

Another way to increase the predominantly mechanical load of a grinding process is to adjust the grinding wheel properties [7]. The bond of the grinding wheel plays a

significant role in the formation of loads in the process and the resulting surface and subsurface characteristics [8]. Grinding wheels with an elastic bonding system are mainly used in finishing processes to achieve high surface finishes and can reliably generate a workpiece roughness of Rz = 1 μm [9]. Wagner analyses the difference between the nominal depth of cut $a_{e,nom}$ and the actual depth of cut $a_{e,real}$. This difference is a factor of 3 to 5, depending on the elasticity of the bonding system [10]. When grinding with an elastic bonding system, a preload is initially built up in grain engagement. Material removal only occurs at relatively high normal forces. These high normal forces lead to residual compressive stresses. In addition to the improved surface topography, these are expected to contribute to an increase in the components' service life [11]. Heymann, Aurich et al. and Kipp et al. [12–14] confirmed an improved wear behaviour when grinding cutting edges with grinding wheels with an elastic bonding system.

*1.3. Further Tool Concept*

Another special tool concept in the field of finishing processes has been investigated by Uhlmann et al., who used abrasive brushes [15]. These create an irregular surface due to their elastic behaviour. The relationship between the achievable roughness and the material removal was investigated by Müller [16] using such tools. It was found that the high compliance combined with low pressure resulted in low material removal rates and improved material surfaces.

*1.4. Coarse Grains*

A special tool concept was also used for machining brittle-hard materials, where high mechanical loads are required for ductile grinding [7]. Grains set in a defined way on grinding wheels, so-called engineered grinding wheels, were investigated for their ability to enable ductile material removal. The work of Koshy et al. simulates the influence of grain protrusion as well as grain shape and arrangement on the achievable roughness [17]. In the model of Denkena et al. [18], these tools were used to investigate the complex interaction mechanisms at the interface between the bonding system and abrasive grain.

García Lunaa et al. investigated the influence of defined set and sharpened grains on developing forces and surface integrity [19]. Defined flattened grains were used to increase the grinding forces. The shape and size of the abrasive grains of grinding tools are also relevant to the achievable surface and subsurface properties. In [20], the positive effect of flattened coarse diamond grinding wheels on the formation of residual compressive stresses in the workpiece of AISI 4140 subsurface is shown. Grain refinement was observed in the subsurface, leading to an increase in hardness and the desired residual compressive stresses. In their investigations, Aurich et al. [21] were able to demonstrate an improvement in the grinding behaviour and process forces of the defined set grinding wheels compared to standard electroplated grinding wheels.

The knowledge gained in grinding so far suggests the potential to increase the mechanical process load by blocky, almost spherical, coarse grains and by means of an elastic bonding system. A combination of the proven positive effects in a tool concept to exploit the mechanical material loading during grinding has not been investigated before and represents the research approach taken here.

**2. Materials and Methods**

The grinding tool concept derived from the state of the art consists of an elastic bonding system made of foamed polyurethane and uses a geometrically defined, almost spherical grain shape instead of conventional abrasive grains with sharp cutting edges. The defined grain shape is a spherical rounded steel wire with a diameter of approximately 800 μm and a hardness of 57 HRC. This spherical grain shape ensures that the engagement conditions are shifted towards micro-ploughing, which increases the mechanical load. The new tool concept compared to conventional tools is shown in Figure 1.

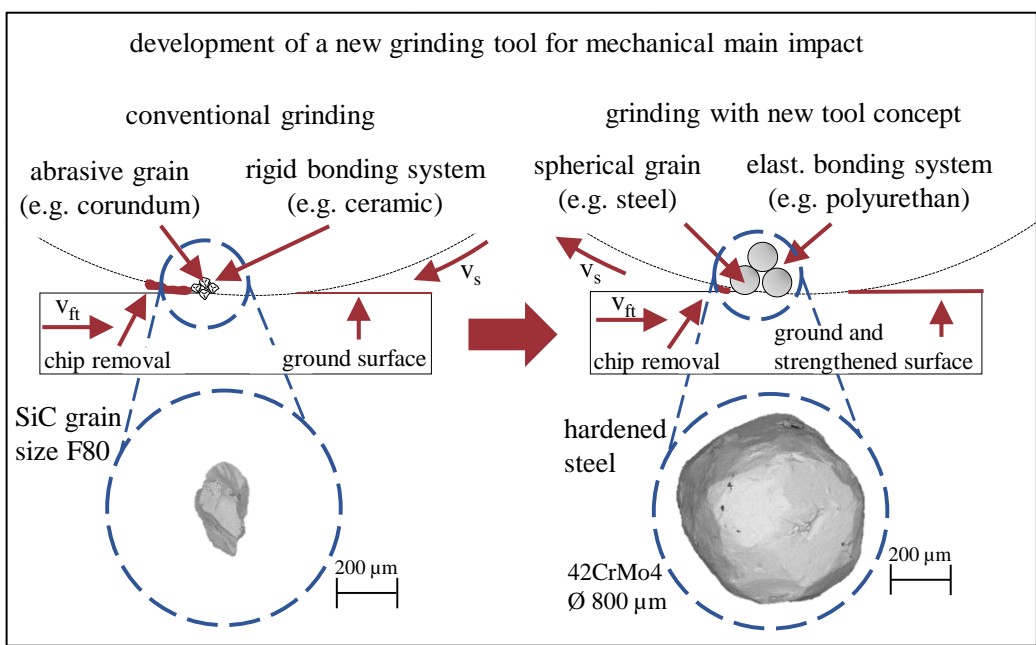

**Figure 1.** Schematic representation of the conventional grinding tool (**left**) and the novel tool concept (**right**).

To investigate the process behaviour, cylindrical specimens of quenched and tempered AISI4140 with a hardness of 47HRC are ground on the Studer S41 grinding machine using the grinding wheel described above in an external cylindrical plunge grinding process (Figure 2). In addition, the chemical composition of the steel used is shown in Table 1.

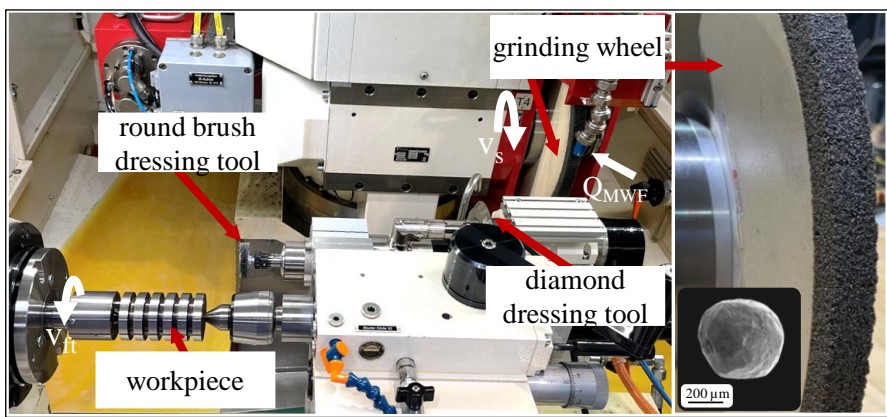

**Figure 2.** Machine workspace and kinematics of OD plunge grinding process.

**Table 1.** Chemical composition of AISI4140.

| Notation Unit | C % | Cr % | Mn % | P % | S % | Si % | Mo % | Ni % | Al % | Cu % |
|---|---|---|---|---|---|---|---|---|---|---|
| AISI 4140 | 0.448 | 1.09 | 0.735 | 0.012 | 0.002 | 0.264 | 0.244 | 0.200 | 0.018 | 0.065 |

In order to maximise the effect of the process, the grinding wheels must be sharpened to achieve a defined protrusion and undamaged shape of the spherical grains. This is achieved by using nylon brushes impregnated with silicon carbide, which reset the elastic bonding system after an initial conventional dressing with a diamond form roller. This dressing strategy is based on an elastic deformation of the brushes along the grains and the consequent removal of the elastic bonding system while leaving the spherical grains intact.

The effect of the new tool on the process behaviour, and thus on the resulting surface and subsurface characteristics and topography, is investigated on the basis of the main influencing variables for external cylindrical plunge grinding. Throughout the experiments, the cooling lubrication was kept constant with an oil flow rate of $Q_{MWF}$ = 60 L/min, as well as the grinding wheel sharpening conditions with a brush cutting depth of $a_{ed}$ = 1 mm, a radial feed rate of $v_{frd}$ = 0.9 mm/min, a process dwell time of $t_{pd}$ = 120 s and a grinding wheel speed equal to the main experiments of $v_{cd}$ = $v_c$ = 22.5 m/s. The pre-grinding process parameters were chosen on the basis of a fine finishing process with grinding wheel speed $v_c$ = 40 m/s, depth of cut $a_e$ = 0.1 and radial feed rate $v_{fr}$ = 0.14 mm/min. The specimens were pre-ground (Tyrolit 1-400 × 10 × 127 A 80 H 9 V) to ensure a reproducible starting condition by means of a finishing process using a conventional vitrified grinding wheel (SiC abrasive grains) and were characterised by a roughness of Rz = 7.5 ± 0.93 µm.

The main influencing parameters of the plunge grinding process are the total depth of cut $a_e$ and the speed ratio q. Due to the elastic behaviour of the tool bonding and the associated deviation between the nominal and actual depth of cut $a_e$, it is also useful to vary the time in which the spherical grains are engaged in the grinding process [22]. Therefore, the process dwell time $t_p$ is varied. The parameters are shown in Figure 3. To consider and evaluate the underlying mechanisms induced by the spherical grain engagements, min-max values for the main influencing parameters are considered. Taking into account the elastic bonding system and the preload required for material removal, the total depth of cut $a_e$ is selected with $a_e$ = 15 µm, $a_e$ = 25 µm and $a_e$ = 40 µm and is thus within the range of average roughness values (Rz = 3–8 µm) according to the state of the art. The speed ratio q is varied within the range of typical finishing processes with q = 91, q = 102 and q = 114. Furthermore, the process dwell time (after reaching the total nominal cutting depth $a_e$) $t_p$ is varied with $t_p$ = 0 s, $t_p$ = 3 s and $t_p$ = 6 s. The radial feed rate was kept constant at $v_{fr}$ = 2.5 mm/min in order to achieve the depth of cut $a_e$ quickly.

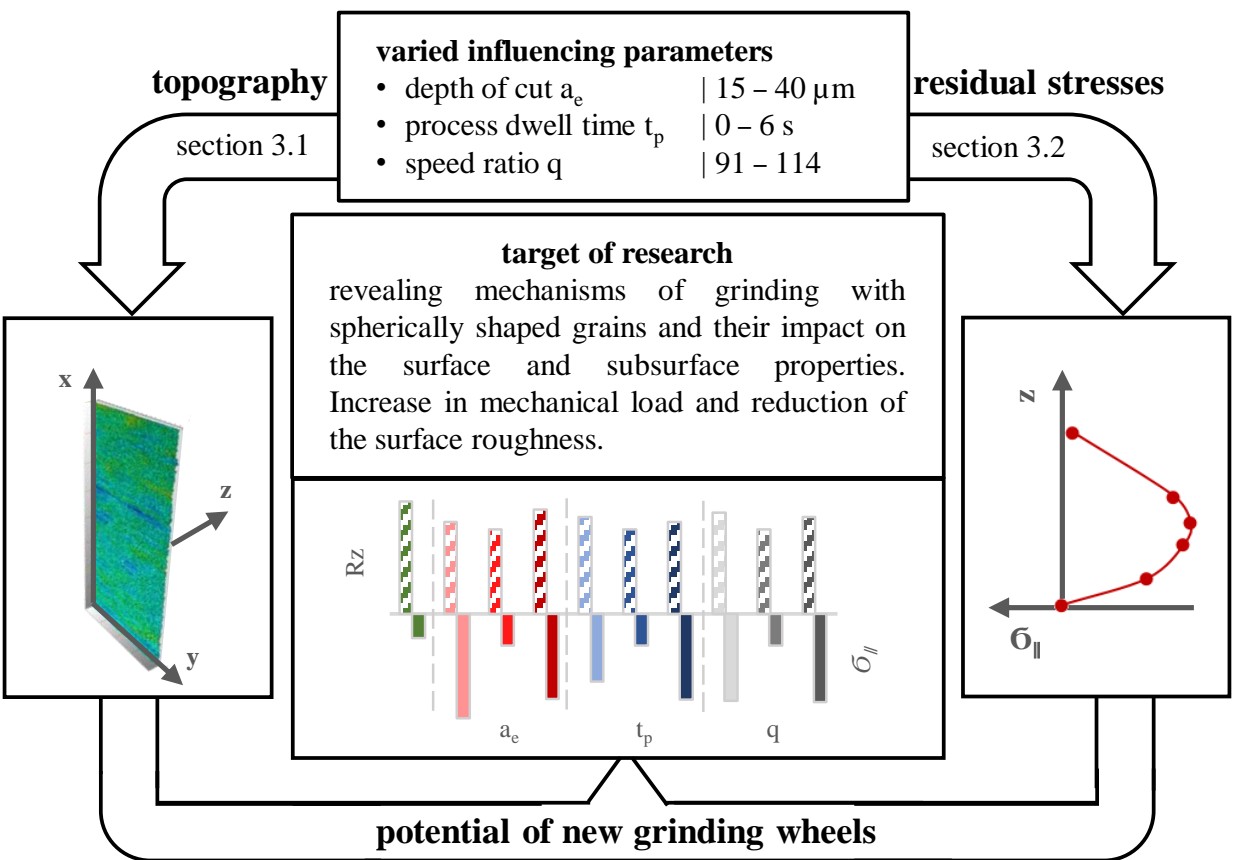

**Figure 3.** Target of research, varied influencing parameters and impact on surface and subsurface.

As a first approximation of the mechanical load in the process, the resulting grinding forces are measured using a Kistler 9366CC 3-component dynamometer. The evaluation and assessment of the mechanical effect was investigated using optical measurements of the topography with the Bruker Alicona InfiniteFocus as well as the force measurements shown in Section 3.1. Further characterisation of the surface and subsurface properties was carried out by metallographic cross sections and residual stress depth profiles measured with the Seifert XRD Charon SXL as described in Section 3.2. Detailed images of the microstructure were taken with Electron Backscatter Diffraction (EBSD) using the Philips XL 30.

## 3. Experimental Results

To investigate the process effect of the new tool in terms of the mechanical load on the surface integrity, the results of the grinding experiments are divided into two subsections according to the forces and the topography on the one hand and subsurface properties on the other. Clear effects on topography and subsurface properties were found when varying the depth of cut $a_e$ and the process dwell time $t_p$.

### 3.1. Mechanical Load and Surface Topography

In order to assess the mechanical load in the machining process, the first step is to determine the normal force generated during the process. For this purpose, the averaged maximum normal force determined from three measurements is assumed to be the relevant influencing variable for work hardening due to an instantaneous modification of the subsurface properties. To help understand how the forces are evaluated, Figure 4 shows an exemplary force curve for the new tools. It is shown in dashed bars, as is the tangential force coupled with the thermal load, shown in solid dashed bars, for the variations in depth of cut $a_e$, process dwell time $t_p$ and speed ratio q in Figures 5–7. In addition, the roughness parameters Ra (left dashed bar) and Rz (right dashed bar) averaged from three measurements/samples are shown for the same variables to give an indication of the local plastic deformation of the roughness peaks during machining.

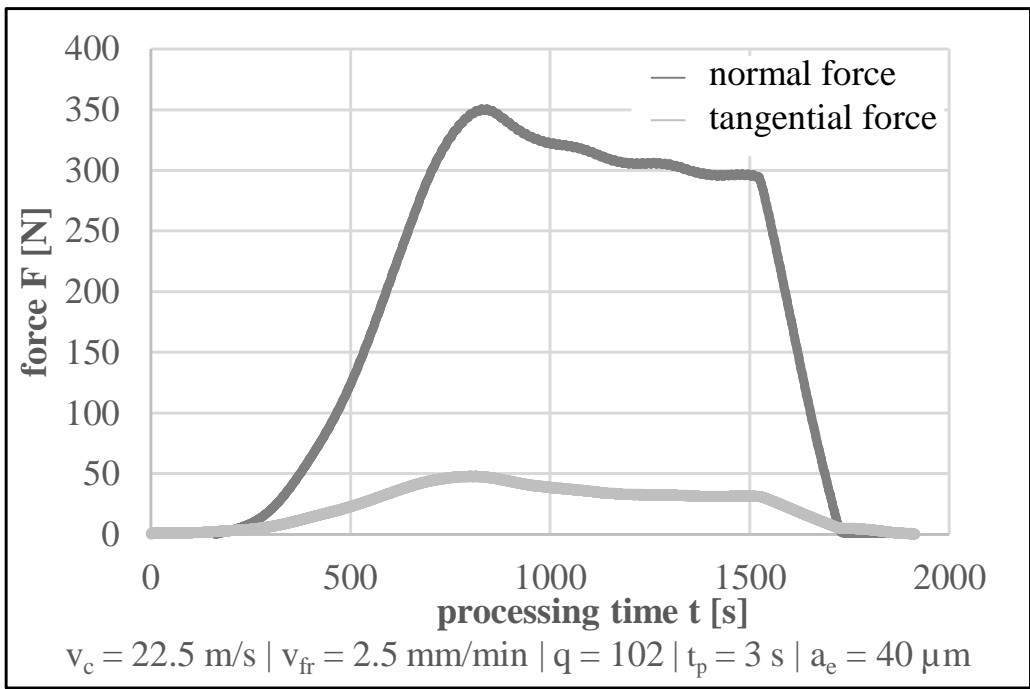

**Figure 4.** Exemplary force curves for the new tools.

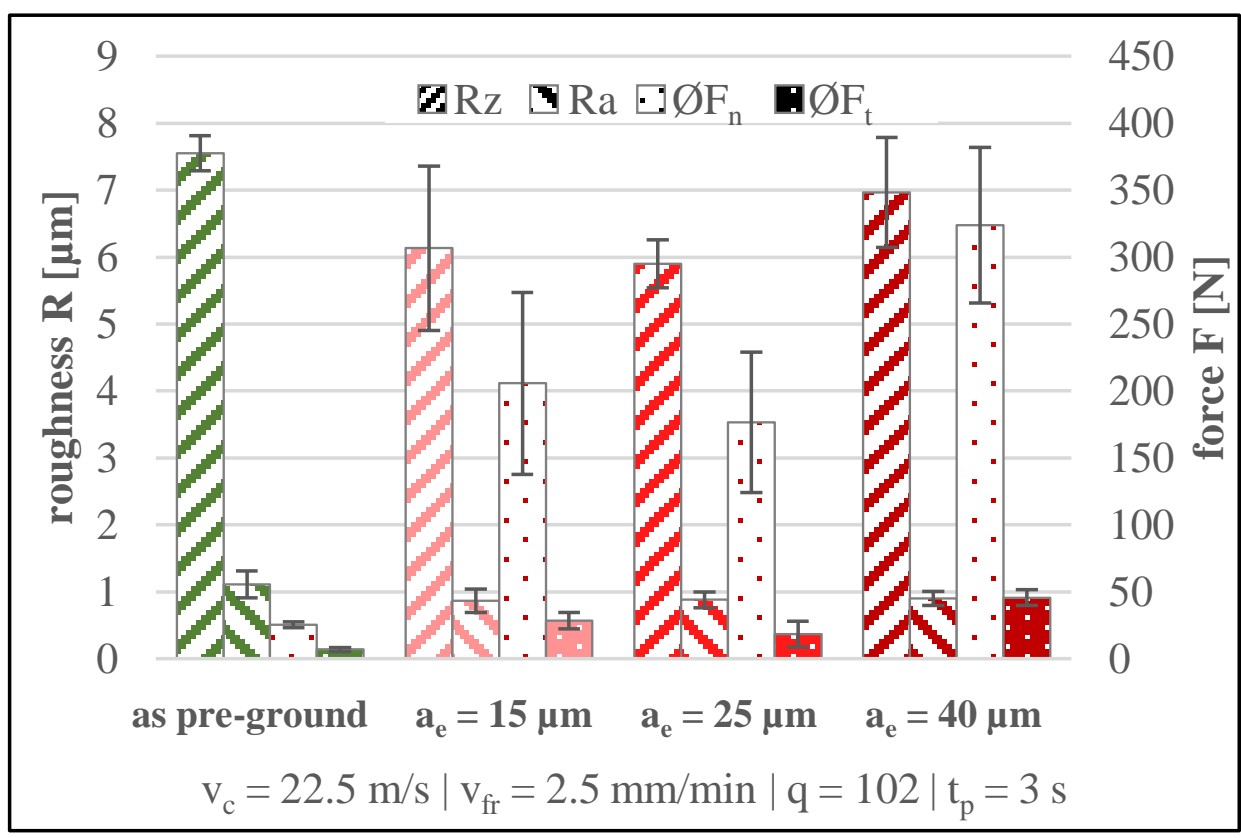

**Figure 5.** Roughness and forces with varied total depth of cut $a_e$.

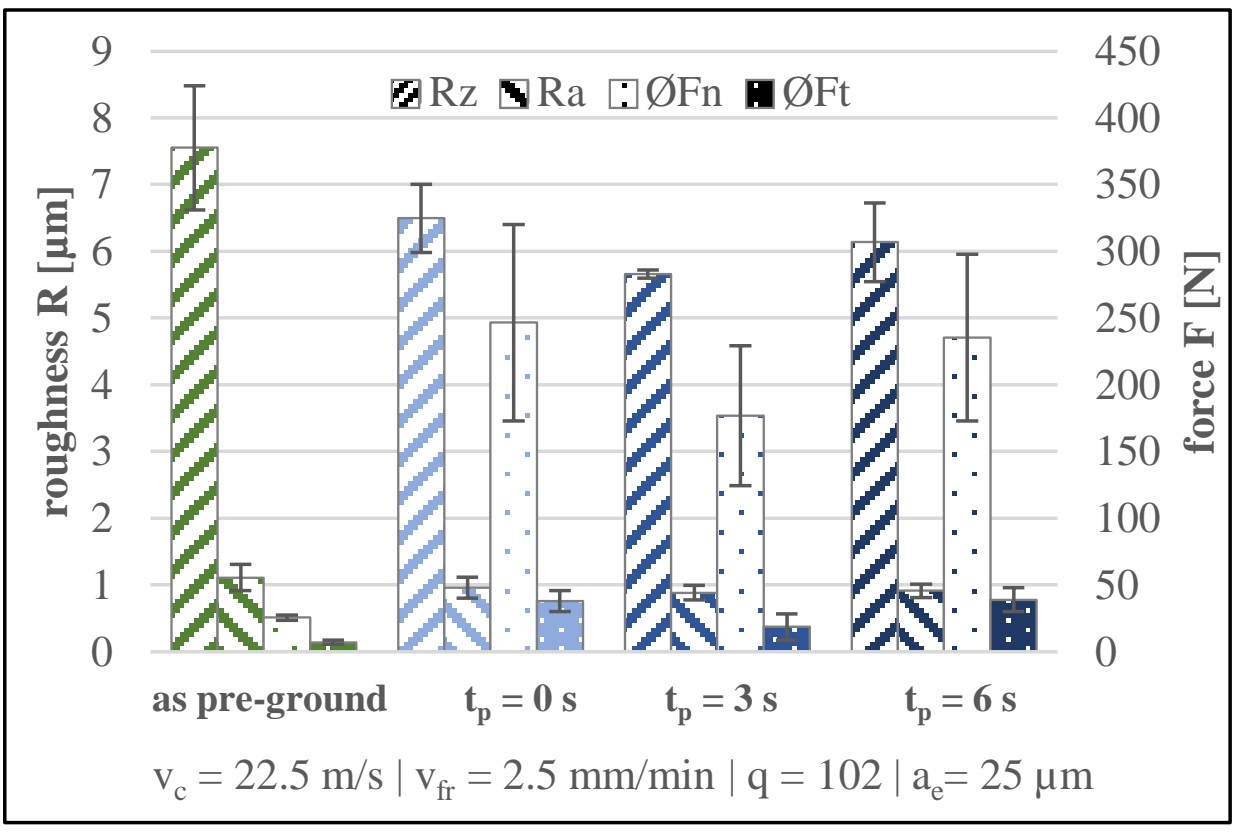

**Figure 6.** Roughness and forces with varied process dwell time $t_P$.

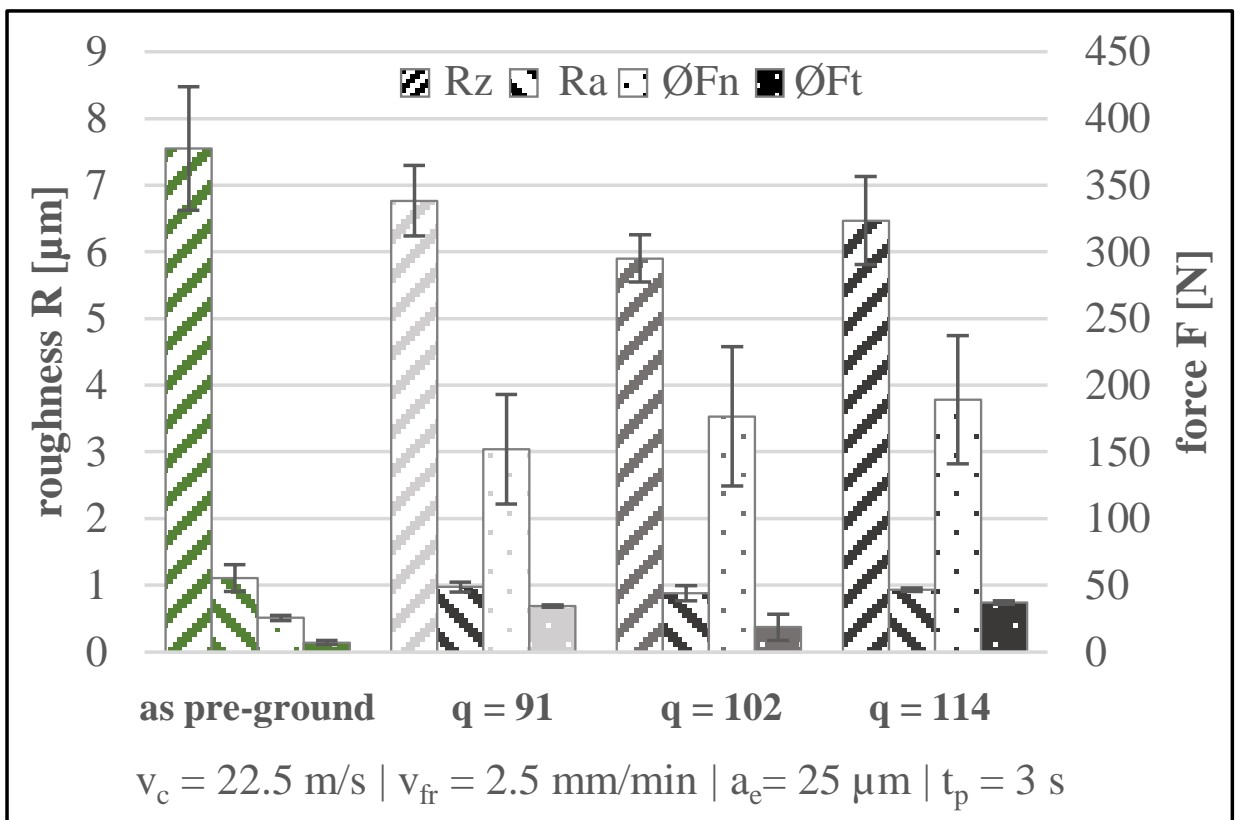

**Figure 7.** Roughness and forces with varied speed ratio q.

This shows an increase in the normal force $F_n$, which is determined by the radial feed rate [23]. The forces reach a steady level when the expansion and the pushing back cancel each other out. This is the point at which the maximum load is applied to the workpiece, which determines the mechanical stress. The tangential force $F_t$ behaves in the same way. The grinding force ratio µ can be used as a measure of the distribution of the thermal and mechanical process effect. It is calculated from the quotient of the tangential force and the normal force µ = Ft/Fn. A reduced grinding force ratio of approx. 0.14 compared to conventional grinding processes with approx. 0.35 is clearly visible [24,25].

The level of normal force $F_n$ is significantly higher for all variations of the influencing parameters compared to those for pre-grinding with a conventional vitrified SiC tool. At the same time, the grinding force ratio remains at a comparable level of 0.14 ± 0.2. Therefore, a significant increase in the mechanical portion of the process load can be assumed. For both values of Ra and Rz, the roughness tends to decrease compared to the conventionally pre-ground condition. The variation of the depth of cut $a_e$ indicates that an increased depth of cut $a_e$ is associated with a significant increase in normal force. The depth of cut $a_e$ appears to be the main variable influencing the normal force and therefore the mechanical load during machining. This is associated with a tendency for the roughness to increase which results from a slightly deeper indentation of the grains in the workpiece surface. An increase in deformation work is also observed at the maximum depth of cut $a_e$.

Considering the process dwell time $t_p$, both the roughness parameters and the normal force remain at approximately the same level. As is typical of grinding wheels with an elastic bonding system, the grinding wheels remove the roughness peaks and create a new surface topography. Similarly, the process dwell time $t_p$ has only a limited effect on the forces shown here as they represent the averaged maximum normal force achieved at the start of the process. However, due to the longer process time $t_p$, more impacts of single grains occur at the surface of the workpiece, which is relevant regarding the subsurface properties.

The investigated speed ratios q were selected based on the values for a finishing process. A direct comparison of the three levels shows a minimal change in the roughness parameters Ra and Rz. There is a tendency for the roughness to decrease as the speed ratio increases. The reason for this is that there are more contacts of the spherical grains per workpiece surface element. The expected assumption of decreasing forces due to an increased number of grains cannot be confirmed. The tangential force shows a constant level while the normal force tends to increase. An explanation for this is the different chip removal behaviour. In a conventional process, material removal is the dominating mechanism, so each individual grain has to remove less material due to the higher relative cutting speed. Due to the significantly higher deformation of the material caused by the spherical grains, each grain performs a higher amount of forming work. The resulting tangential force is caused by the friction and shearing of the material, both supposed to increasing with increasing relative cutting speed.

In order to gain a better understanding of the roughness parameters, 3D measurements of the workpiece topography were performed. Figure 8 shows the reference condition after pre-grinding and the average parameter set during grinding with the novel wheel. This shows an improvement in the surface finish due to the use of spherical steel grains. The new grinding wheel flattens the distinctive grooves caused by the pre-grinding tools. It can be assumed that during the process, the roughness peaks are pressed into the roughness valleys. The overall result is a more homogeneous and flattened workpiece surface.

| workpiece 42CrMo4 | | external cylindrical plunge grinding up-grinding | |
|---|---|---|---|
| pre-grinding | | new tool concept | |
| $a_e = 0.1$ mm | $v_s = 40$ m/s | $a_e = 50$ µm | $v_s = 22.5$ m/s |
| $n_w = 140$ U/min | $v_{fr} = 0.14$ mm/min | $n_w = 70$ U/min | $v_{fr} = 2.5$ mm/min |
| $t_p = 0$ s | | $t_p = 3$s | |

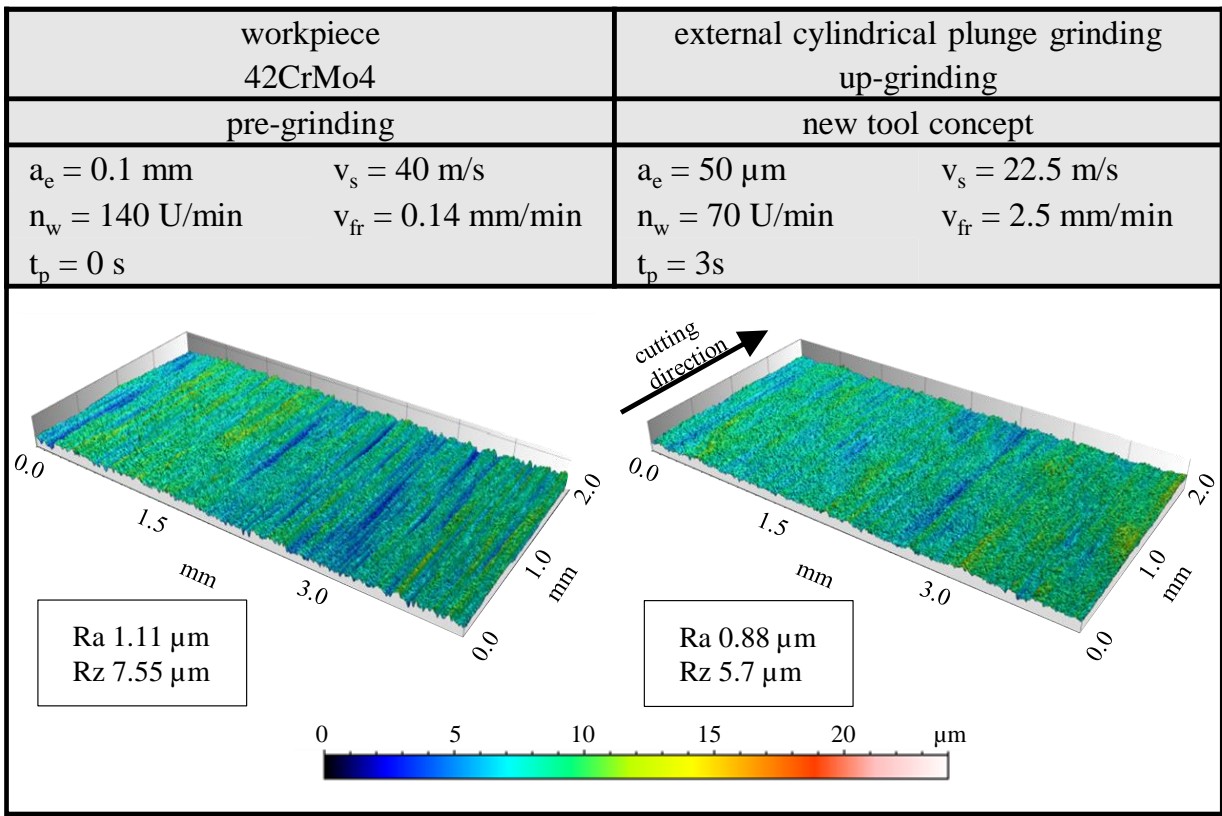

**Figure 8.** Surface topographies after pre-grinding with a conventional tool (**left**) vs. after sub-sequent grinding with the new grinding tool concept (**right**).

As an intermediate summary, it can be stated that the mechanical load can be adjusted varying the depth of cut $a_e$, but at the same time, an increase in the machining time $t_p$ as well as the speed ratio q has only a small effect on the roughness and the forces. The previously defined objective of improving surface roughness by using grinding wheels with an elastic bonding system was achieved.

The resulting roughness values with the new tool concept are characterised by a certain robustness with respect to the variations in $a_e$, $t_p$ and q in the investigated parameter range. For the industrial application, this can be regarded as an advantage, and in turn, the thorough selection of the wheel specification is of importance.

However, forces and roughness are only a part of the relevant resulting process and work result characteristics, so a more detailed analysis of the subsurface characteristics is carried out.

### 3.2. Impact on Subsurface Characteristics

The consideration of forces appears to be primarily an indication of the mechanical load and impact mechanisms due to grain engagement. In the following, further quantities are used to characterise the mechanical impact of the load on the surface and subsurface based on residual stresses and micrographic cross sections. Analogous to the results presented in Section 3.1, the residual stresses and cross sections are presented separately.

As described in Section 2, the Seifert XRD Charon SXL is used to measure the inherent stresses. This produces X-rays of the Cr-K$\alpha$ type. Lattice strain is measured using the sin2$\psi$ method on the {211} plane of the material. For the variation of the main influencing parameter, depth of cut $a_e$, the residual stresses parallel to $v_c$ are shown in Figure 9 and the micrographic cross sections in Figure 10a–c and an EBSD image in Figure 10d.

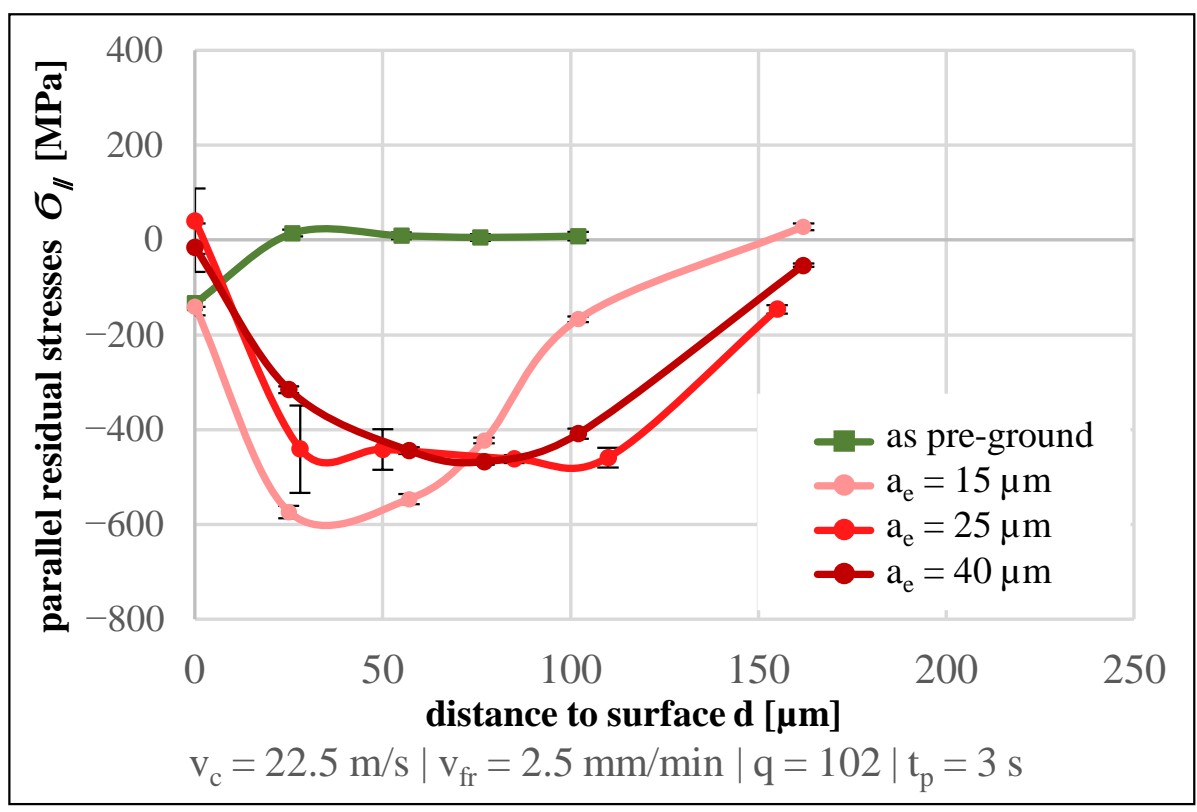

**Figure 9.** Residual stress profiles for varied depth of cut $a_e$.

Corresponding to the normal forces (Figure 5), an increased mechanical load as a result of the increased depth of cut $a_e$ is shown in Figure 9. The typical residual stress depth profile for pre-grinding with a maximum impact at the surface associated with a low depth effect differs significantly from stress profiles obtained with the novel grinding tool. Significantly increased compressive residual stress level and a significant increase in depth effect below the surface are observed. The maximum compressive residual stress is located below the surface, as it is commonly observed for various mechanical surface treatment processes [6,26]. As the depth of cut $a_e$ increases, the maximum residual stress is shifted towards higher depths and the absolute value of the maximum is reduced. It

can be assumed that the increased friction due to deeper grain engagement generates a higher temperature, which counteracts the mechanical stress and limits the formation of compressive residual stress in the area close to the surface.

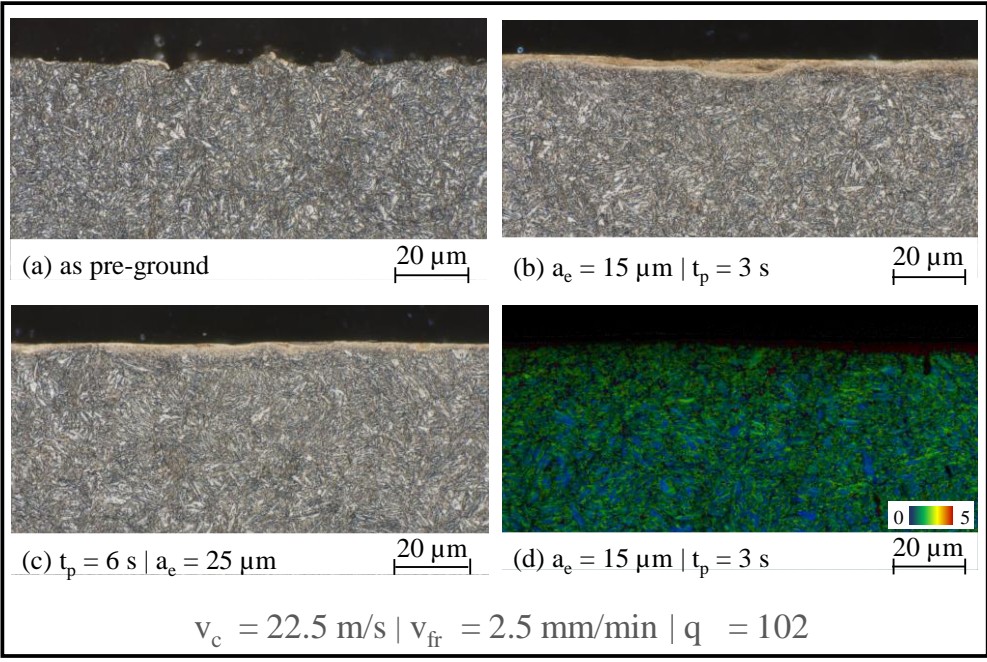

**Figure 10.** Micrographic cross sections (**a**–**c**) and EBSD measurement (kernel average misorientation map 0–5°) (**d**).

Along with the microstructural cross sections shown in Figure 10b, a bright layer is visible in the near surface region, indicating a nanocrystalline microstructure due to the high shear stresses induced [27,28]. In addition, an oriented grain texture is obtained at the depth of the residual stress maxima, which is associated with an increase in dislocations. At this point, a significant temperature rise that would lead to the formation of a white layer on the surface can also be ruled out. The dark colouring of the underlying structure due to annealing effects, which occurs in conjunction with the white layer, cannot be seen in the microstructural cross sections [29]. The EBSD image shown in Figure 10d indicates a reduction in grain size in the area close to the surface, as well as a darker coloured area, which correlates with plastic deformation as indicated by the kernel average misorientation in the EBSD analysis, confirming the increasing mechanical load in the surface and subsurface area.

When considering the process dwell time $t_p$, the results again confirm that the determination of the forces is not sufficient to characterise the mechanical load and the associated effect. The increasing number of contacts as a result of the increased process dwell time provides an increase in the compressive residual stress maximum while the depth effect remains almost constant (Figure 11). An exception is the process dwell time of 6 s, which shows a lower depth effect. Increased impact of the spherical grains can potentially lead to softening of the workpiece surface, which is seen here in the early stages. The effect of process dwell time for the novel tool is shown to be similar in importance to that of spark out in conventional grinding [30]. The micrograph in Figure 10c again shows the formation of a bright layer, which is indicative of a nanocrystalline microstructure close to the surface [27,28].

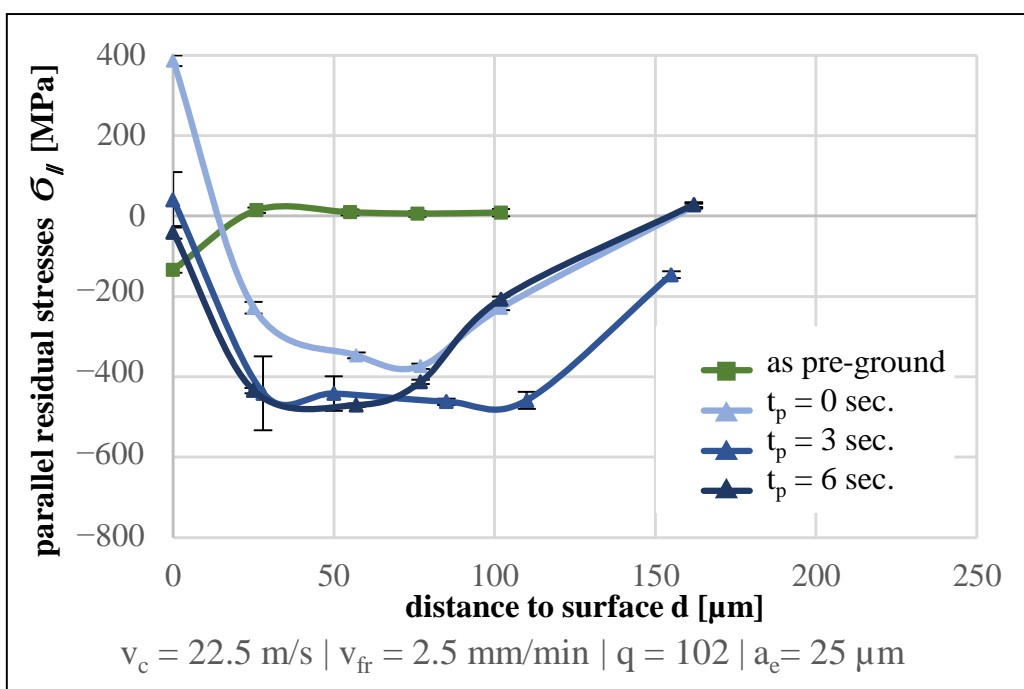

**Figure 11.** Residual stress depth profiles for varied process dwell time $t_P$.

Figure 12 shows the effect of the variation of the speed ratio q. Apparently, the depth effect increases with increasing speed ratio q, which could be related to an increase in grain contacts per workpiece surface element, analogous to the process dwell time $t_P$. The maximum residual stress remains almost constant over the range studied.

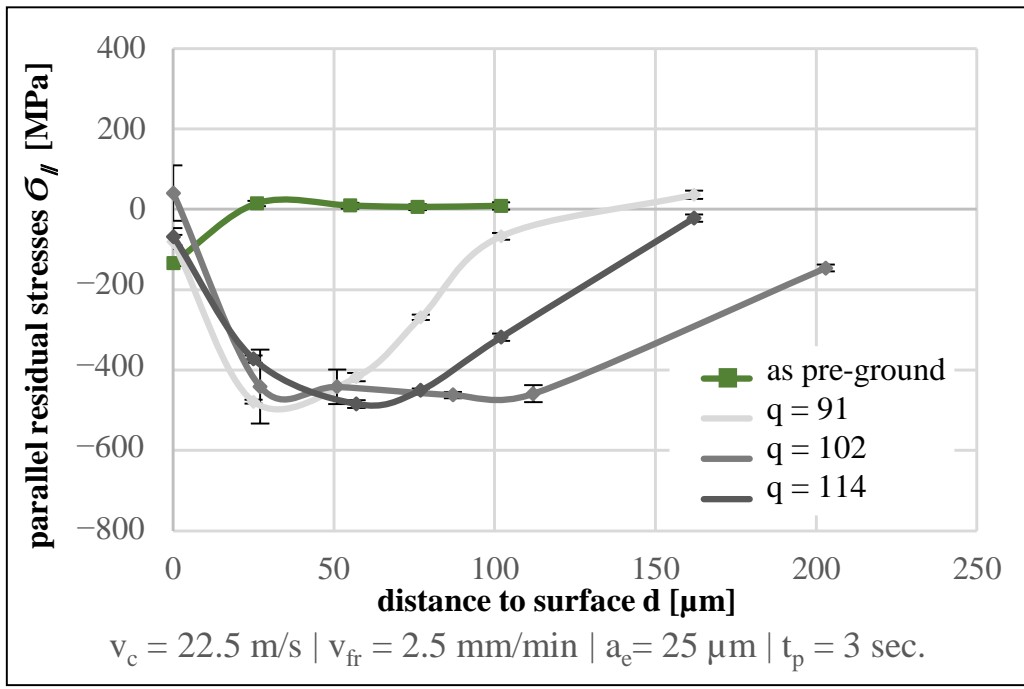

**Figure 12.** Residual stress depth profiles for varied speed ratio q.

## 4. Conclusions

The use of the newly designed grinding wheel with a spherical grain makes it possible to increase the mechanical load during machining. As a result, significant compressive residual stresses are induced with maxima below the workpiece surface. The maximum

compressive residual stresses tend to decrease with increasing depth of cut $a_e$, but the depth effect increases slightly. For both, dwell time $t_p$ and speed ratio q, the depth effect of compressive residual stresses increases significantly due to the higher number of contacts. On the other hand, the effect on the maximum compressive residual stresses is negligible. Effects described for different processes of mechanical surface treatment [26] appear to be transferable into a grinding machine environment through the new tool concept. Based on this, savings in set-up and non-productive times should be achieved through a process chain within a machining centre by using different grinding wheels, which means an integration of the mechanical surface treatment into a grinding machine by application of the novel tool concept for the finishing step.

In terms of roughness, the spherical grains generally lead to an improvement. In this context, an increase in depth of cut $a_e$ results in an increase in roughness, while the speed ratio q and the dwell time $t_p$ lead to an improvement due to the increasing number of grain engagements. In future investigations, further variations of the grinding wheels should be investigated by varying the Young's modulus of the bonding system as well as alternative spherical grain materials, e.g., ceramics. In addition, variation of the spherical grain concentration and spherical diameter may be of interest as influencing parameters. In order to achieve better roughness values, an approach can also be taken in which the grinding wheels are made up of two parts: one part with a conventional vitrified bonded abrasive and the other part with the new tool concept presented in the paper.

**Author Contributions:** Conceptualisation, M.E., C.H. and D.M.; investigation, visualisation, methodology, formal analysis and writing—original draft preparation, M.E.; writing—review and editing, M.E., D.M. and C.H.; project administration, C.H. and D.M. All authors have read and agreed to the published version of the manuscript.

**Funding:** The scientific work has been funded by the Deutsche Forschungsgemeinschaft (DFG) within the collaborative and transregional research centre "Process Signatures"—project number 223500200—(SFB/TRR136 in Aachen, Bremen & Oklahoma), sub project T07 (INST 144/531-1).

**Data Availability Statement:** The data presented in this study are available on request from the corresponding author.

**Acknowledgments:** The authors express their sincere thanks to the DFG for funding this project.

**Conflicts of Interest:** The authors declare no conflict of interest.

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
