# Peer review of "Utilisation Potential of Mechanical Material Loads during Grinding by Means of a Novel Tool Concept"

_jmmp, doi:10.3390/jmmp7050167_

Round 1
Reviewer 1 Report
1. Figures 4, 5, 6, 7 - please make corrections. There should be no unit labels on the numerical axes. Units should be indicated in the axis labels.
2. Lines 183-184: Please provide a source or results from your own research. What is the grinding force ratio in the conventional grinding process?
3. What was the chemical composition of the ground material?
4. Lines 219-220: What could be the reason for the lack of decreasing force with an increased number of grains?
5. Lines 238-239 state that the goal was to improve roughness, but Figure 3 indicates a different goal. Please clarify or indicate that a different goal is being referred to.
6. Please specify how the residual stresses were calculated.
Author Response
Dear Reviewer,
We greatly appreciate the comments on our work by the reviewers which have resulted in its improvement. Additionally to the revised text, please find below the authors’ response to each point raised by the reviewers.
With kind regards,
M. Eich on behalf of the co-authors
- Figures 4, 5, 6, 7 - please make corrections. There should be no unit labels on the numerical axes. Units should be indicated in the axis labels.
A: The adjustments have been made to the units on the axes.
- Lines 183-184: Please provide a source or results from your own research. What is the grinding force ratio in the conventional grinding process?
A: Two sources have been added showing the grinding force ratio in a conventional grinding process. This is approximately at a value of 0.35. It is calculated from the quotient of the tangential force and the normal force µ = Ft/Fn.
- What was the chemical composition of the ground material?
A: The chemical composition of the ground material was added.
- Lines 219-220: What could be the reason for the lack of decreasing force with an increased number of grains?
A: An explanation for the lack of the expected decreasing tangential force was added in the manuscript.
An explanation for this is the different chip removal behaviour. In a conventional process, material removal is the dominating mechanism, so each individual grain has to remove less material due to the higher relative cutting speed. Due to the significantly higher deformation of the material caused by the spherical grains, each grain performs a higher amount of forming work. The resulting tangential force is caused by the friction and shearing of the material, both supposed to increasing with increasing relative cutting speed.
- Lines 238-239 state that the goal was to improve roughness, but Figure 3 indicates a different goal. Please clarify or indicate that a different goal is being referred to.
A: The deviation between the set goals was clarified in figure 3.
- Please specify how the residual stresses were calculated
A: The residual stresses were not calculated. The stresses were measured with the Seifert XRD Charon SXL using the sin2ψ method on the {211} plane of the material. Information on the XRD analysis was added to the manuscript.
Reviewer 2 Report
The authors presented a new grinding tool concept featuring nearly spherical grains in an elastic bonding system for improving the surface and subsurface properties of steel parts, as well as uncovered the underlying mechanisms leading to the intended improvement of surface integrity. The research results would be interested readers in industrial field and academic field, also the grinding machine manufacturers. The manuscript could be accepted after the following issues were addressed.
1. The picture of new grinding tool should be provided more than just schematics.
2. In line 183 and 184, how the ratio 0.14 calculated?
3. Grinding forces are measured using a Kistler Z21487A1 3-component dynamometer, which I think is not accurate. Is there alternative that suitable to measuring the grinding force directly?
Author Response
Dear Reviewer,
We greatly appreciate the comments on our work by the reviewers which have resulted in its improvement. Additionally to the revised text, please find below the authors’ response to each point raised by the reviewers.
With kind regards,
M. Eich on behalf of the co-authors
- The picture of new grinding tool should be provided more than just schematics.
A: An image of the grinding tools was added in figure 2.
- In line 183 and 184, how the ratio 0.14 calculated?
A: The grinding force ratio is calculated from the quotient of the tangential force and the normal force µ = Ft/Fn.
- Grinding forces are measured using a Kistler Type 9255C 3-component dynamometer, which I think is not accurate. Is there alternative that suitable to measuring the grinding force directly?
A: Thank you for pointing this out. There was an error in the designation of the measuring system. A Kistler type 9366CC 3-component dynamometer was used. This has been corrected in the manuscript.